# High *ME1* Expression Is a Molecular Predictor of Post-Transplant Survival of Patients with Acute Myeloid Leukemia

**DOI:** 10.3390/cancers15010296

**Published:** 2022-12-31

**Authors:** César Alexander Ortiz Rojas, Abel Costa-Neto, Diego A. Pereira-Martins, Duy Minh Le, Dominique Sternadt, Isabel Weinhäuser, Gerwin Huls, Jan Jacob Schuringa, Eduardo Magalhães Rego

**Affiliations:** 1Department of Medical Images, Hematology and Clinical Oncology, Medical School of Ribeirao Preto, University of São Paulo, Ribeirao Preto 14048-900, Brazil; 2Center for Cell Based Therapy, University of São Paulo, Ribeirao Preto 14051-060, Brazil; 3Hematology Division, LIM31, Medical School, University of São Paulo, São Paulo 05403-000, Brazil; 4D’Or Institute for Research and Education (IDOR), São Paulo 04501-000, Brazil; 5Department of Experimental Hematology, Cancer Research Centre Groningen, University Medical Centre Groningen, University of Groningen, 9700 RB Groningen, The Netherlands

**Keywords:** acute myeloid leukemia, hematopoietic stem cell transplantation, *ME1*, prognosis, biomarker

## Abstract

**Simple Summary:**

Acute myeloid leukemia (AML) is a blood cancer caused by genetic aberrations acquired by bone marrow progenitor cells, impeding healthy hematopoiesis. While AML is a heterogenous disease and variable parameters can impact AML prognosis, the options for treatments remain limited. The first line of treatment continues to be chemotherapy, usually followed by a hematopoietic stem cell transplant (HSCT) obtained from a compatible healthy donor. Of those transplanted patients, only about 50–60% will be long-term survivors. Consequently, the identification of markers that may predict the resulting HSCT outcome is a medical need. To address this issue, we applied different mathematical models at diagnosis to the transcriptome of AML patients who were treated with standard chemotherapy and then subjected to HSCT, in order to uncover genes associated with the clinical outcome post-transplant. By doing so we identified the *ME1* gene, whereby high expression of *ME1* was associated with worse prognosis. Furthermore, *ME1* expression was correlated with energetic processes related to oxidative phosphorylation. Our study reveals that *ME1* is an important biomarker and a potential therapeutic target.

**Abstract:**

Several laboratory and clinical variables have been reported to be associated with the outcome of intensive chemotherapy for acute myeloid leukemia (AML), but only a few have been tested in the context of hematopoietic stem cell transplant (HSCT). This study aimed to identify genes whose expression of AML at diagnosis were associated with survival after HSCT. For this purpose, three publicly available adult AML cohorts (TCGA, BeatAML, and HOVON), whose patients were treated with intensive chemotherapy and then subjected to allogeneic or autologous HSCT, were included in this study. After whole transcriptome analysis, we identified *ME1* as the only gene whose high expression was associated with shorter survival in patients subjected to HSCT. In addition, the inclusion of *ME1* expression was able to improve the European LeukemiaNet risk stratification. Pathways related to lipid biosynthesis, mainly fatty acids, and cholesterol were positively correlated with *ME1* expression. Furthermore, *ME1* expression was associated with an M2 macrophage-enriched microenvironment, mature AML blasts hierarchy, and oxidative phosphorylation metabolism. Therefore, *ME1* expression can be used as biomarker of poor response to HSCT in AML.

## 1. Introduction

Allogeneic (allo-HSCT) and autologous (auto-HSCT) hematopoietic stem cell transplantation (HSCT) are potential curative treatments for acute myeloid leukemia (AML) [1,2]. The most used transplant in AML is allogeneic HSCT, which is reported to reduce the risk of relapse by more than 60% when performed in the first phase of complete remission [3]. However, undesirable events such as graft failure, graft versus host disease, and relapse can occur and contribute to mortality [4]. Additionally, disease and donor characteristics are associated with the outcome of allo-HSCT. As a result, the European Leukemia Net (ELN) 2017 (ELN2017) classification distinguishes three risk groups with significantly distinct prognoses in patients with AML receiving allo-HSCT [5,6]. Furthermore, the allo-HSCT seems to be a better option for high and intermediate-risk patients [3,7,8], while favorable and intermediate-risk patients presented promising results in clinical trials when treated with chemotherapy alone or followed by auto-HSCT [9,10]. Despite this progress in AML treatment, refinement of the ELN classification is needed [6,8]. In this context, several biomarkers have been proposed as predictors of survival in patients treated with HSCT [11], such as measurable residual disease (MRD) quantification based on the expression of certain genes, e.g., mutated *NPM1* and *WT1* or the expression of *BAALC* and *MN1*. Another option is the identification of aberrant or asynchronous immunophenotypes [5,12,13], or the stearic acid/palmitic acid ratio on day 7 after transplant [14]. Additionally, the combination of the ELN2017 with early post-transplant MRD assessment refines the prediction of relapse after allo-HSCT [5,15].

Herein, we performed a transcriptome analysis of three independent cohorts of adult patients with AML to identify genes associated with survival after HSCT. We found that Malic Enzyme 1 (*ME1*) gene expression levels in mononuclear cells of diagnostic samples were associated with survival after HSCT. In addition, we demonstrated that *ME1* potentially refines the ELN risk categorization for HSCT-treated patients, while also being correlated with a tumor-supportive microenvironment, non-GMP-like AML blasts subtype, and specific AML drug sensitivity.

## 2. Materials and Methods

### 2.1. Patients and Gene Expression Profiling

Publicly available clinical and transcriptomic data of three adult AML cohorts whose patients were treated with intensive chemotherapy and then subjected to allogeneic or autologous HSCT were included in this study. RNA-sequencing data from two cohorts were included: The Cancer Genome Atlas (TCGA, *n* = 69 patients), with 63 and 6 patients treated with allo- and auto-HSCT, respectively [16]; and the BeatAML (*n* = 77), in which all patients were subjected to allo-HSCT [17]. These data series were sequenced on Illumina HiSeq2000 and HiSeq2500, respectively. Normalized gene expression and clinical data for the TCGA cohort were retrieved from the Firebrowse data portal (www.firebrowse.org, accessed on 1 August 2021), while for the BeatAML cohort, we used the Beat AML Vizome data portal (www.vizome.org, accessed on 1 July 2021) and the cBioPortal for Cancer Genomics (www.cbioportal.org, accessed on 1 July 2021). Data concerning the HOVON cohort (*n* = 134) consisted of microarray gene expression analysis performed using an Affymetrix Human Genome U133 Plus 2.0 Array [18,19]. In this cohort, 86 and 48 patients were treated with allo- and auto-HSCT, respectively, according to the Dutch–Belgian Hematology–Oncology Cooperative Group (HOVON) trial. Normalized gene expression data were retrieved from the Gene Expression Omnibus (GEO) database (GSE6891 dataset) (www.ncbi.nlm.nih.gov/geo/, accessed on 1 July 2021). To compare the gene expression between mononuclear cells from AML patients and normal hematopoietic stem/progenitor cells, gene-expression data were retrieved from the Bloodspot database (http://servers.binf.ku.dk/bloodspot/, accessed on 1 July 2021).

To evaluate the impact of *ME1* expression in the prognosis of AML, we retrospectively included a cohort composed of 80 consecutive patients with newly diagnosed AML, followed from January 2003 to December 2021 at the University Medical Center Groningen (UMCG). Of those, 32 patients corresponded to *de novo* AML, received intensive chemotherapy, and were included for validation studies. Twenty of the patients received HSCT as consolidation therapy. Gene expression of *ME1* was analyzed by RT-qPCR using the primers forward: 5′-CAACAGTCAGGAGATCCAGGT-3′ and reverse: 5′-GGAGATCCATTAAGAGAAGATACCTG-3′. The RNA derived from bone marrow mononuclear cells (BMMC) was isolated using density gradients. The gene expression values of *ME1* were calculated as relative quantification using the ∆Ct method. In compliance with the Declaration of Helsinki and with the institutional review board-approved protocol (#NL43844.042.13), informed consent was obtained from all patients. All AML subtypes were considered eligible for study, apart from patients with acute promyelocytic leukemia. Functional assays performed on AML samples included in the UMCG cohort (flow cytometry evaluation for macrophage profiling and ex vivo drug screenings/metabolic features) were performed as previously published [20,21].

### 2.2. Searching Strategy and Derivation of Prognosis Associated Genes

The cohorts were dichotomized into groups with high and low gene expression. This was carried out after calculating the optimal cut point value for each gene using the receiver operating characteristic (ROC) curve for censored overall survival data. Then, univariate and multivariate Cox proportional hazard regression was computed. According to data availability, we adjusted prognosis prediction for confounders as follows: age (as continuous variable), sex (male vs. female), white blood cell counts (WBC, as continuous), type of transplant (allo-HSCT vs. auto-HSCT), and European LeukemiaNet categorization (ELN2010 for TCGA and HOVON; ELN2017 for BeatAML). Additionally, we considered only genes that generated Area Under the Curve (AUC) > 0.5, with sensitivity (sens) and specificity (spe) > 0.5. These analyses were performed for each gene of the transcriptome in each cohort. Then, genes consistently associated with prognosis were further investigated after merging the results of the three cohorts. Kaplan–Meier (KM) plots were performed to demonstrate stratification power. In addition, disease-free survival (DFS) and event-free survival (EFS) were analyzed in the TCGA and HOVON cohorts, respectively, according to data availability.

### 2.3. Biological Pathways, Cells Signatures, and Drug Sensitivity Prediction

To identify the biological roles of the genes associated with HSCT outcomes, we evaluate the protein interactions using the STRING webtool (https://string-db.org/, accessed on 1 September 2021). In addition, we explored the biological pathways associated with these genes by running the Gene Set Enrichment Analysis (GSEA) using the Broad Institute software [22]. The Gene Ontology (GO), Kyoto Encyclopedia of Genes and Genomes (KEGG), and Reactome databases were used in our analysis. Enrichment scores were calculated based on Kolmogorov–Smirnov statistics and tested for significance using 1000 permutations. Additionally, Spearman correlation was used as a metric for ranking genes. A minimum gene size of five was set for gene sets. A pathway was considered enriched when the nominal *p*-value and false discovery rate (FDR) *q*-value were <0.05. Next, with the aim of predicting the tumor microenvironment complexity according to the genes associated with survival, we evaluated the cell signatures of the AML samples using the CIBERSORT [23], EPIC [24], xCell [25], MCP-counter [26], quanTIseq [27], and TIMER2.0 [28] algorithms from the TIMER2.0 webtool (http://timer.cistrome.org/, accessed on 1 September 2021). Additionally, we applied the deconvolution approach proposed by Zeng et al., 2022 [29] to infer the abundance of seven leukemic cell types among non-tumor cells. For that purpose, we applied the CIBERSORTx algorithm (https://cibersortx.stanford.edu/, accessed on 1 September 2021) [30] executed in absolute mode and applying S-mode batch correction using the single cell signature matrix and single cell reference sample available in https://github.com/andygxzeng/AMLHierarchies (accessed on 1 September 2021). After deconvolution, we normalized the malignant cell populations to 1. Then, we predicted the four AML subtypes proposed by Zeng et al., 2022 [29] after clustering the AML patients based on their leukemia cell hierarchies. Finally, we decided to explore the correlation between drug response and the expression levels of the genes of interest. To do so, we used the AUC values resulting from 157 and 159 drugs tested ex vivo by Tyner et al., 2018 [17] and Lee et al., 2020 [31], respectively. Additionally, we evaluated 448 drugs tested in AML cell lines. AUC values were available in the Genomics of Drug Sensitivity in Cancer (GDSC) resource (https://www.cancerrxgene.org, accessed on 1 September 2021) [32]. The following AML cell lines were considered: CMK, GDM-1, HEL, HL-60, KASUMI-1, KG-1, ME-1, MOLM-13, MOLM-16, MONO-MAC-6, NOMO-1, OCI-AML2, OCI-AML3, OCI-AML5, OCI-M1, P31-FUJ, PL-21, SIG-M5, and THP-1. Gene expression data of cell lines were retrieved from the Expression Atlas database (www.ebi.ac.uk/gxa/experiments/E-MTAB-2770/, accessed on 1 October 2021). Spearman rho coefficients > 0.3 or <−0.3 with *p* < 0.05 were used to identify strong correlations.

### 2.4. Cell Lines and Drugs

All cell lines and in vitro cultures were maintained in a humidified atmosphere at 37 °C with 5% CO_2_. For the cell lines, mycoplasma contamination was routinely tested to assure mycoplasma-free cultures. All leukemia cell lines were authenticated by short tandem repeat analysis. The HL60 (CCL-240™) cells were obtained from the American Type Culture Collection and grown in IMDM (for HL60; Gibco, Eindhoven, NL) with 10% FCS. The MOLM13 (ACC 554) and TF1 (ACC 334) cell lines were obtained from the DSMZ-German Collection of Microorganisms and Cell Cultures and grown in RPMI-1640 (Gibco, NL) with 10% FCS + 1% sodium pyruvate. TF1 cells were cultured with 10 ng/mL of IL-3 (Sandoz, Almere, NL). Decitabine (DEC), 2-deoxy-glucose (2DG), and Midostaurin (PKC) were obtained from Sigma-Aldrich (St. Louis, MO, USA) and cytarabine (citarax) was obtained from Blau pharmaceuticals (São Paulo, Brazil). Venetoclax (VEN) was obtained from Selleckchem (Houston, TX, USA). Ruxolitinib was obtained from MedChemExpress (Groningen, NL).

### 2.5. Lentiviral Vectors and Lentivirus Production

For ME1 knockdown, two sequences for shME1 (shME1#1—TRCN0000064732 and shME1#2—TRCN0000064730; Sigma-Aldrich, St. Louis, MO, USA) vectors were properly expanded and validated as previously described [33]. A shRNA sequence that does not target human genes (referred to as scrambled) was used as a control (hereinafter called shCTRL—generated as previously described [33]).

### 2.6. In Vitro Assays

#### 2.6.1. Apoptosis Assay

For the apoptosis analysis, 5 × 10^4^ cells were seeded in 48-well plates and incubated in complete medium for 72 h, in the presence of vehicle and the above-described drugs. The apoptosis rate was determined using the Annexin V-APC and 4′,6-diamidino-2-phenylindole (DAPI) binding assay (BD Biosciences, San Jose, CA, USA) in the presence of calcium buffer. All specimens were acquired by flow cytometry using the FACS LSR-II machine (Becton-Dickison, Franklin Lakes, NJ, USA) and analyzed with the FlowJo software v.10 (Treestar, Inc., Ashland, OR, USA). All experiments were performed at least in triplicate, and for each sample a minimum of 10,000 single cell events were acquired.

#### 2.6.2. Oxygen Consumption (OCR) and Extracellular Acidification Rate (ECAR) Measurements

Extracellular flux measurements were performed using a seahorse XF96 analyzer (Agilent, Santa Clara, CA, USA) as described by the manufacturer. Briefly, the oxygen consumption rate (OCR) and Extra Cellular Acidification Rate (ECAR) were measured using the Seahorse XF96 analyzer (Seahorse Bioscience, Agilent, US) at 37 °C. For AML cell lines, 1 × 10^5^ viable cells (evaluated by DAPI-using flow cytometry measurements) were seeded per well in poly-L-lysine (Sigma-Aldrich) coated (incubation at room temperature for 20 min) Seahorse XF96 plates in 180 μL XF Assay Medium (Modified DMEM, Seahorse Bioscience), respectively, in the presence of 2 mM of Glutamine and NaOH to adjust the final pH. For OCR measurements, XF Assay Medium was supplemented with 10 mM Glucose and 2.5 µM oligomycin A (Port A), 2.5 µM FCCP (carbonyl cyanide-4-(trifluorometh oxy) phenylhydrazone) (Port B), and 2 µM antimycin A, together with 2 µM Rotenone (Port C), were sequentially injected in 20 µL volume to measure basal and maximal OCR levels (all reagents from Sigma-Aldrich). For ECAR measurements, Glucose-free XF Assay medium was added to the cells in addition to 10 mM Glucose (Port A), 2.5 µM oligomycin A (Port B), and 100 mM 2-deoxy-D-glucose (Port C) (all reagents from Sigma-Aldrich). All XF96 protocols consisted of 4 times mix (2 min) and measurement (2 min) cycles, allowing for determination of OCR at basal and in between injections. Both basal and maximal OCR levels were calculated by assessing metabolic response of the cells in accordance with the manufacturer’s suggestions. The OCR measurements were normalized to the viable number of cells used for the assay.

#### 2.6.3. T-Cell Mediated Cytotoxicity Assay

Cytotoxicity of T cells against AML cell lines displaying different levels of ME1 gene expression was performed as described elsewhere [34]. Briefly, target cells (T) containing stable expression of shME1#1/shME1#2 and shCT, were incubated with cell trace violet (ThermoFisher, Waltham, MA, USA) stained T cells at 1:1, 1:2, and 1:5 effector-to-target (E:T) ratios. Targets in media alone were used for assessment of spontaneous death. After 72 h of culture, viable cell counts (defined as Annexin-V-Viability marker-cells; Zombie NIR was used as a viability marker—Biolegend, USA) was measured. The mean percentage of specific lysis of technical duplicate samples was calculated as 100* (spontaneous death + experimental death)/(spontaneous death—background). A total of 4 independent T cell donors were used for these experiments. Cells were acquired using NovoCyte Quanteon Flow Cytometer Systems (Agilent, Santa Clara, CA, USA), and a minimum of 1000 viable cells per condition were acquired per sample.

### 2.7. Statistical Analysis

Overall survival (OS) was defined as the time from diagnosis to death from any cause. Disease-free survival and event-free survival (DFS/EFS) were defined as the time from CR achievement to the first adverse event; that is, relapse or death from any cause, whichever occurred first. To adjust *ME1* expression for confounding factors, we performed a multivariate Cox proportional hazard regression. Kaplan–Meier (KM) plots were performed to demonstrate stratification power. The Mann–Whitney test and Fisher’s exact test were used to compare clinical and biological characteristics between patients. All *p*-values were two-sided, with a significance level of 0.05. All calculations were performed using R 4.1.1 (The CRAN project) software.

## 3. Results

### 3.1. ME1 Gene Was Identified as an Independent Factor to Predict the Survival of AML Patients Subjected to HSCT

To identify robust prognostic gene expression biomarkers for AML patients treated with intensive chemotherapy and then subjected to allogeneic or autologous HSCT, we evaluated clinical and transcriptomic data of three publicly available adult AML cohorts: TCGA, BeatAML, and HOVON. The cohorts were dichotomized according to gene expression by using the ROC curve for survival. Then, the high and low expression categories were established for each gene in each cohort to evaluate their impact on prognosis (Figure 1A). Surprisingly, we identified only one gene, named *ME1*, whose high expression was associated with shorter overall survival in all cohorts (Figure 1B, Appendix A). Kaplan–Meier plots showing the power of stratification of *ME1* expression for overall survival are shown in Figure 1C. AML patients with high *ME1* expression had shorter survival compared to those with low expression (median OS of 21.5 vs. 53.9 months for TCGA, 18.9 months vs. not reached for BeatAML, and 18.4 vs. 145.9 months for HOVON). In addition, high *ME1* expression levels were associated with shorter DFS (TCGA) and EFS (HOVON) (median DFS of 10.2 vs. 20.6 months and median EFS of 12.9 vs. not reached) (Figure 1D). Furthermore, we compared the *ME1* expression between healthy hematopoietic stem cells (HSCs; Lin^−^, CD34^+^, CD38^−^, CD90^+^, CD45RA^−^), common myeloid progenitors (CMPs; Lin^−^, CD34^+^, CD38^+^, CD45RA^−^, CD123^+^), or granulocyte-macrophage progenitors (GMPs; Lin^−^, CD34^+^, CD38^+^, CD45RA^+^, CD123^+^), with mononuclear cells from AML patients. We detected that *ME1* expression was comparable between leukemic cells and HSCs, but higher in AML cells, when compared with healthy CMPs and GMPs (Figure 1E). Appendix A depict the main clinical and laboratory characteristics of each cohort, and the association of these variables with the expression level of *ME1*.

As shown in Figure 2A–C, after multivariate Cox regression analysis, *ME1* expression remained an independent factor for overall survival, with a HR of 1.87 (*p* = 0.045) for the TCGA, 3.74 (*p* = 0.039) for the BeatAML, and 2.08 (*p* = 0.003) for the HOVON cohort. In addition, AUC values were higher than 0.5 for patients in the high-*ME1* expression group during the first 3 years of follow-up in all cohorts (Figure 2D). Next, we evaluated whether *ME1* expression could improve the prognostic power of ELN risk groups. As shown in Figure 3A,E, the risk stratification of ELN2017 (Log-rank, *p* = 0.33) was refined after the inclusion of *ME1* expression (Log-rank, *p* = 0.014). In addition, when we compared the impact of *ME1* expression on survival of patients treated with HSCT vs. chemotherapy, we found that *ME1* was a stronger risk gene in patients treated with HSCT than with chemotherapy alone. Similarly, survival predictions of *ME1* were inconsistent when applied to all AML cases irrespective of the treatment type (Figure 3F). In agreement with this result, we verified whether the disease status at the end of induction was associated with high or low *ME1* expression, which could affect the outcome of HSCT. As shown in Appendix A, there was no link between *ME1* expression and complete response (CR/CRi) or refractory case distribution. Next, we quantified *ME1* expression by RT-qPCR in our own cohort (UMCG), consisting of 32 *de novo* AML patients treated with standard chemotherapy as induction therapy. *ME1* expression was associated with short survival rates (Log-rank, *p* = 0.046) (Appendix A). Twenty out of thirty-two patients received HSCT as consolidation therapy; of these, six died after transplantation, and five cases presented high *ME1* expression (35.7 vs. 16.7%).

### 3.2. ME1 Is Associated with an Immunosuppressive TME, Mono/cDC-like Leukemia, and Increased Sensitivity to a Specific Set of Drugs

Previous studies have described *ME1* to be a cytosolic protein that catalyzes the conversion of malate to pyruvate while concomitantly generating NADPH from NADP [35]. Similar results were obtained in our STRING analysis (Figure 4A,B). Next, our gene set enrichment analysis identified 853 and 304 biological pathways that were positively and negatively enriched, respectively (Figure 4C). Notably, IL-10 and IL-1 signaling pathways were positively enriched (Figure 4D), while processes such as RNA splicing, rRNA, and tRNA processing pathways were negatively enriched in patients with high *ME1* expression (Figure 4E). Pathways related to the regulation of ROS production and lipid biosynthesis and metabolism, as well as processes critically modulated by NADPH disposition, were positively correlated with *ME1* expression (Appendix A). Additional pathways related to cellular stress and metabolism were consistently enriched in all three cohorts (Appendix A). Next, since IL-1 and IL-10 are mediators of polarization of innate immune cells, we evaluated which immune and stromal cell signatures were associated with *ME1*^high^ leukemic blast cells in all three cohorts. We found that AML cells with high *ME1* expression have a bone marrow microenvironment infiltrated by M2 macrophages (Figure 5A), which was confirmed by flow cytometry in our validation cohort (UMCG). The frequency of monocytic/macrophage cells (defined as SSC^high^CD45^high^HLA-DR^+^) presenting the M2 marker, CD36, in AML primary samples was higher in patients with high *ME1* expression (Figure 5B). In addition, after performing a deconvolution analysis to reveal the hierarchical cellular composition of AML samples (Appendix A), we found a high abundance of non-tumor monocytes signature (Appendix A). Interestingly, we also found that *ME1* expression was associated with high mono-like and cDC-like AML burden, i.e., a mature AML phenotype, with a low count of GMP-like clones and without differences in leukemic stem/progenitor clone’s abundance (Figure 5C,D).

Next, we determined the drug sensitivity of *ME1*^high^AML cells in an ex vivo drug screen. Our results indicated that *ME1*^high^ cells were more resistant to standard AML chemotherapeutic agents such as cytarabine and daunorubicin, as well as to Bcl-2 family inhibitors (obatoclax and venetoclax) (Figure 6A). In addition, AML cells with high *ME1* expression showed increased sensitivity to several mTOR inhibitors, among other kinase inhibitors. Drug sensitivity in AML cell lines showed that high *ME1* expression was correlated with resistance to histone deacetylase inhibitors (HDACi) and to ruxolitinib (JAK1/2 inhibitor), while being sensitive to CDK9_5038, a CDK9 inhibitor, and ERK_6604, an ERK1/2 inhibitor (Appendix A). To confirm these findings, we silenced the *ME1* gene in TF1, MOLM13, and HL60 cell lines (Figure 6B). Our results showed that the knock-down of *ME1* reduced the proliferative capacity of AML cell lines, demonstrating the impact of *ME1* on key processes contributing to leukemia progression (Figure 6C). Next, we tested the drug-induced apoptosis by standard chemotherapeutic agents and target therapies used in AML on silenced *ME1* cells, and confirmed increased sensitivity of sh*ME1* cells to Bcl-2 inhibitors (VEN) and ruxolitinib (Figure 6D–F). Additionally, we observed an increased sensitivity to cytarabine (AraC) and decitabine (DEC in MOLM-13 and HL-60 cells) when *ME1* was silenced. We also noted an inverse correlation between *ME1* expression and decitabine-induced apoptosis in our set of ex vivo treated primary AML cells (Appendix A).

Our ex vivo evaluation of primary AML blasts revealed that patients with high *ME1* levels displayed increased mitochondrial metabolism, as demonstrated by increased mitochondrial membrane potential (MMP) levels and corroborated by the positive correlation between *ME1* expression and the basal oxygen consumption (OCR) rate of blasts (Figure 5B). In line with this observation, we also noted that *ME1*^high^ AMLs were more sensitive to complex I inhibitors (rotenone and metformin) and displayed decreased levels of mitochondrial membrane potential in their T cells (as indicated by the term in the left side—Lymphocytic MMP) (Figure 5B). Altogether, these findings suggested that *ME1*^high^ AMLs rely more on oxidative phosphorylation (OXPHOS) for their energy. In addition, *ME1*^high^ AMLs could be associated with immunosuppressive features by modulating T lymphocyte activity. To test these hypotheses, we evaluated the extracellular flux rate of shME1 cells compared to their control (shCT) in a seahorse experiment. We found that *ME1* silencing strongly impacted mitochondrial respiration, since the basal and maximum oxygen consumption rates (OCR) of sh*ME1* cells were clearly reduced (Appendix A). Contrarily, the glycolytic state remained stable in all three cell lines (Appendix A). Finally, to test whether differential *ME1* levels impact the susceptibility of AML cells to T-cell mediated cytotoxicity, we performed co-culture experiments using different T-cell donors and AML cell lines transduced with shME1 and shCT. No differences were observed for TF1 cells expressing differential *ME1* levels, and only one sh*ME1* clone of HL60 cells presented a modest decrease in cell viability upon T-cell exposure (Appendix A). Overall, these results suggest that *ME1* levels do not directly affect the sensitivity of AML cells to T-cells, suggesting that additional factors might play a role in the *ME1*-associated survival prediction of AML patients subjected to HSCT.

## 4. Discussion

In the present study, we show that *ME1* expression is a predictor of post-transplant survival in AML. *ME1* gene belongs to a short family of malic enzyme genes (*ME1*, *ME2*, and *ME3*) and encodes for a cytosolic enzyme that replenishes pyruvate from malate, a process that generates NADPH [36,37]. As described by Pavlova et al. [36], pyruvate is preferentially used for mitochondrial acetyl-CoA generation in quiescent tumor cells, while in proliferating cells, the main use of reduced carbon is reserved for biosynthesis. Here, our analysis indicates that *ME1* expression is positively associated with both neutral lipids’ biosynthesis, mainly fatty acids and cholesterol, and leukocyte proliferation capacity (Appendix A), suggesting that AML cells with high *ME1* expression have a higher rate of proliferation. Corroborating this notion, the proliferation rate of three AML cell lines was reduced upon *ME1* silencing. In addition, our drug sensitivity test indicated that AML cells with high *ME1* expression are resistant to standard chemotherapy drugs and sensitive to kinase inhibitors.

The malic enzymes have been recognized as the main cytosolic NADPH producers, along with isocitrate dehydrogenase (IDH) enzymes and the pentose phosphate pathway [38]. In addition to its critical role as a coenzyme in biosynthesis [36], NADPH is relevant for redox homeostasis, protecting the tumor cells from increased reactive oxygen species (ROS) production during cell proliferation [36]. Thus, NADPH can counteract ROS via the reconstitution of reduced glutathione and thioredoxin proteins [36]. In line with these previous studies, our results indicated an association between *ME1* and the regulation of the stress-related pathways, including ROS process metabolism (Appendix A). Although we did not explore the direct regulation of intracellular NADPH concentration and *ME1* expression, previous studies had already dipped into this issue. In human osteosarcoma U2OS cells and normal diploid fibroblast IMR90 cells, the forced expression of *ME1* or *ME2*, or the addition of a malic enzyme substrate, increased the cellular NADPH levels. The silencing of *ME1* or *ME2* consequently reduced the NADPH levels [39]. Shao et al. [40] demonstrated that cancer cells preferentially depend on *ME1*-mediated production of NADPH to support uncontrolled growth compared with healthy cells. However, they also found an adaptability of cancer cells to *ME1* silencing after long culture periods [40]. Here, we identify that the silencing of *ME1* compromised mitochondrial respiration of AML cells, but not glycolysis, demonstrating that *ME1* plays an active role during the replenishing of substrates for mitochondrial respiration.

Another possible explanation for the prognostic impact of ME1 in HSCT for AML is its association with the *TP53* pathway, which was mainly reported in solid tumors [41,42,43]. Downregulation of *ME1* or *ME2* increases ROS levels because of p53 activation, leading to a strong induction of senescence [39]. In IMR90 and U2OS cells, the knockdown of *TP53* led to a significant increase in basal mRNA levels of *ME1* and *ME2* compared to p53-wild-type cells [39]. In addition, when these cells were treated with genotoxic agents such as etoposide and doxorubicin, a reduction in the expression of *ME1* and *ME2* was shown. However, when *TP53* was knocked down, the expression of these genes no longer responded to DNA damage [39]. Jiang et al. showed that the induced expression of *ME1* or *ME2* in U2OS cells diminished the activation of p53 induced by DNA damage, thereby enhancing tumor cell growth [39]. We did not find differences between *ME1* expression according to *TP53* status in primary AML samples, although this analysis was limited by the number of cases included in this study. Additionally, we did not observe differences in *ME1* expression between AML cell lines with different *TP53* mutational status (Appendix A). We cannot rule out that additional mutated genes, such as *KRAS*, could interact with the *ME1* function, since mutations in the latter were shown to increase *ME1* mRNA levels in colorectal tumors [44,45].

Recently, by using single-cell RNA sequencing-based deconvolution methods, it was demonstrated that the specific cellular hierarchy composition of AML is associated with functional, genomic, and clinical properties, in addition to being associated with response to chemotherapy or targeted therapies [29]. Based on this analysis, four distinct AML subtypes were defined: primitive (LSPC-enriched), mature (enriched for mature Mono-like and cDC-like blasts), GMP (dominated by GMP-like blasts), and intermediate (balanced distribution) [29]. By applying the same method, we found that AML cells with high *ME1* expression are associated with mature AML phenotype and with low GMP-like subtype burden [29]. In line with our findings, we observed that AML cells with high *ME1* expression are more resistant to venetoclax and obatoclax, and more sensitive to mTOR inhibitors, which was also reported by Zeng et al., 2022 when analyzing primitive and mature blasts, respectively [29]. Furthermore, previous studies also reported monocytic AMLs to be more resistant to venetoclax plus azacytidine schemes [46]. Next, our cell type signature analysis also indicated increased infiltration of M2 macrophages in the leukemic microenvironment, which has been associated with a poor clinical outcome in AML [20]. Within this context, the top positive enriched pathway in AML cells with high *ME1* expression was IL-10 signaling, a cytokine with potent anti-inflammatory properties described to promote M2 macrophage polarization [46] and stemness of AML cells [47,48]. Overall, AML blast cells with high *ME1* expression are more mature AML-subtypes, resistant to standard chemotherapies and associated with an enriched M2 macrophage microenvironment.

Several cytogenetic, molecular, and immunological markers are predictors of the outcome of AML upon first diagnosis [5,48,49,50], but only a few are predictive of outcomes after allo-HSCT [51,52,53,54]. Higher levels of CD105 expression; mutations in *WT1*, *RUNX1*, and *TP53*; and the partial tandem duplication of *MLL*, detected at diagnosis, have been identified as predictors of short survival after HSCT [52]. Herein, our analysis of the entire transcriptome of samples at diagnosis obtained from AML patients who had undergone HSCT identified that *ME1* is associated with survival rates in three independent cohorts. The exact mechanism of how *ME1* can impact the HSCT response in AML was not evaluated here, but our data highlight the relevance of the microenvironment components, such as M2-macrophages, and the metabolic state of AML cells.

## 5. Conclusions

In summary, high expression of *ME1* is a poor prognostic marker for AML patients who have been treated with intensive chemotherapy and then subjected to HSCT. Lipid metabolism, non-GMP-like AML blasts hierarchy, microenvironment-enriched M2 macrophages, and resistance to standard chemotherapy agents characterize the AML cells with high *ME1* expression, possibly explaining their association with poor outcomes in AML-HSCT.

## Figures and Tables

**Figure 1 cancers-15-00296-f001:**
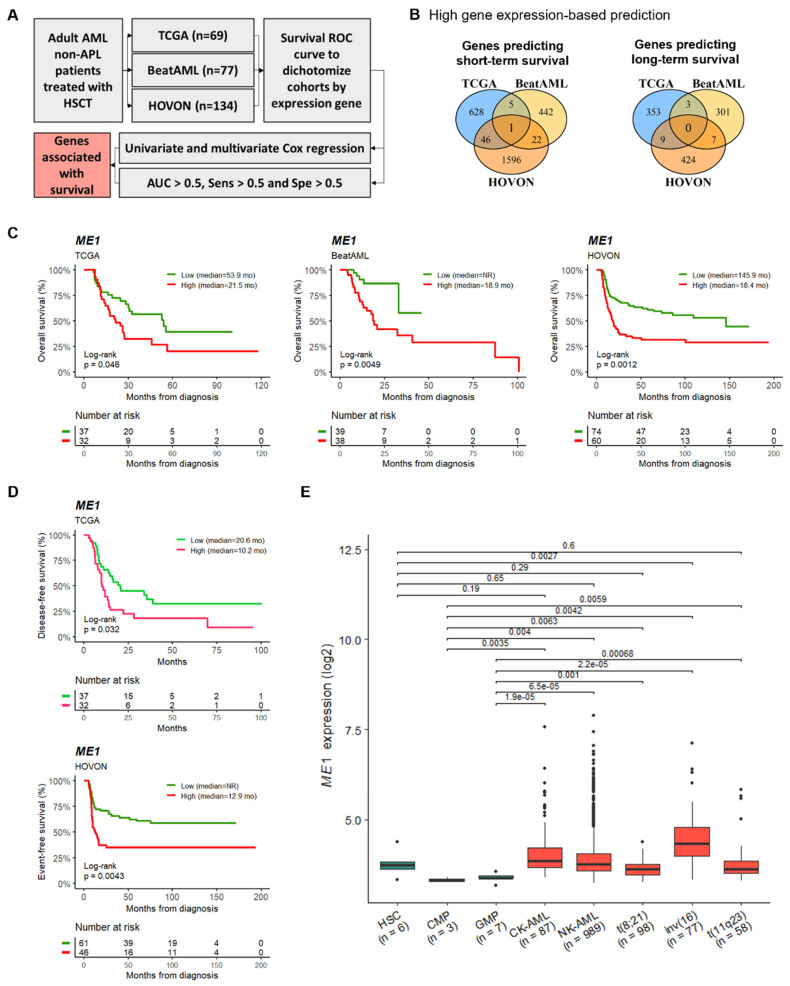
Identification of *ME1* expression as a prognostic marker in HSCT AML. (**A**) Strategy to discover genes associated with survival in HSCT AML. (**B**) Number of genes predicting survival. (**C**) Overall survival (OS), (**D**) disease-free survival (DFS) and event-free survival (EFS), stratified according to *ME1* expression. (**E**) Comparison of *ME1* expression between AML cells and healthy hematopoietic stem/progenitor cells. NR: median survival not reached.

**Figure 2 cancers-15-00296-f002:**
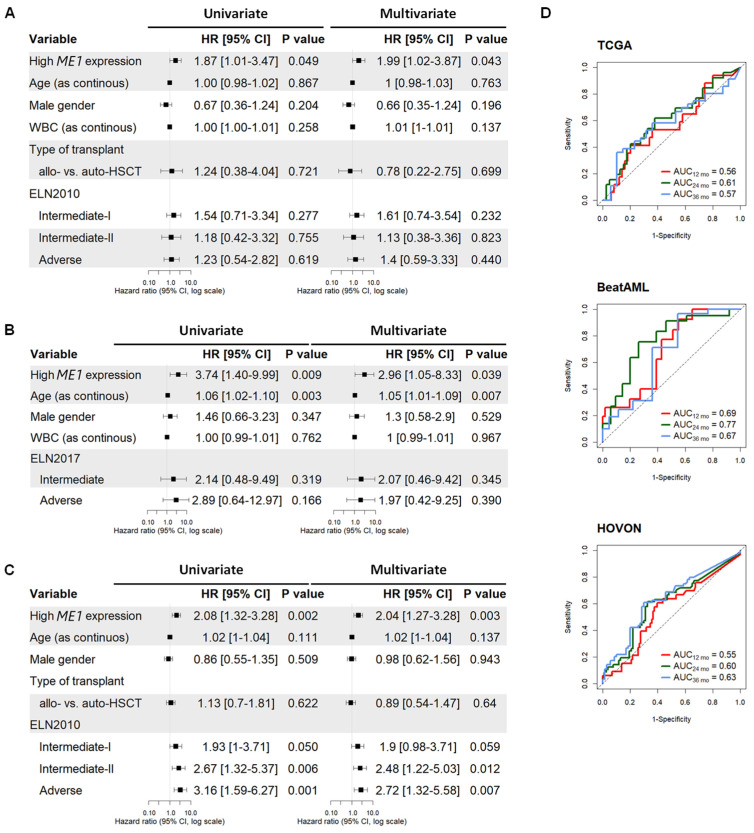
Cox regression and AUC analysis for *ME1* expression in HSCT AML. Univariate and multivariate analysis for (**A**) TCGA, (**B**) BeatAML, and (**C**) HOVON. (**D**) AUC values indicate associations between *ME1* expression and OS.

**Figure 3 cancers-15-00296-f003:**
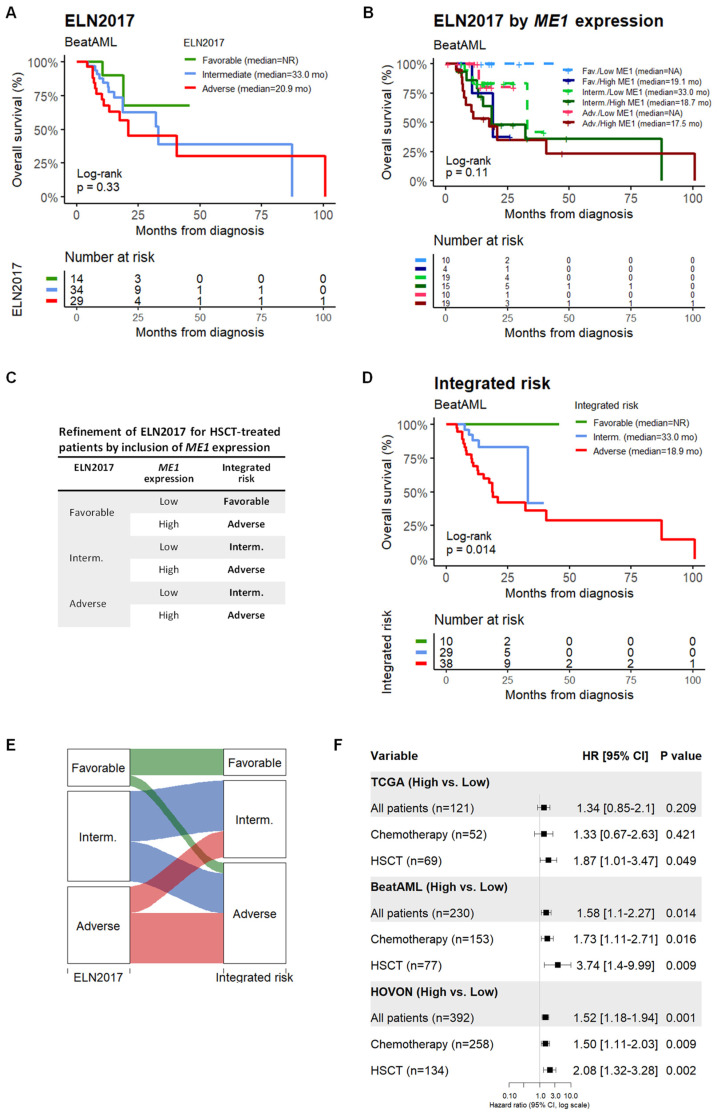
*ME1* expression improves ELN categorization in HSCT AML patients. (**A**) OS resulting from ELN2017 categories. (**B**) ME1 expression stratification by ELN2017 categories. (**C**) Strategy to improve the prognosis prediction of ELN including ME1 expression. (**D**) KM plot showing the improvement of the OS stratification after *ME1* inclusion. (**E**) Recategorization of ELN groups as a consequence of *ME1* inclusion. (**F**) Comparison of death risk by *ME1* expression in patients treated with chemotherapy alone or HSCT in consolidation therapy.

**Figure 4 cancers-15-00296-f004:**
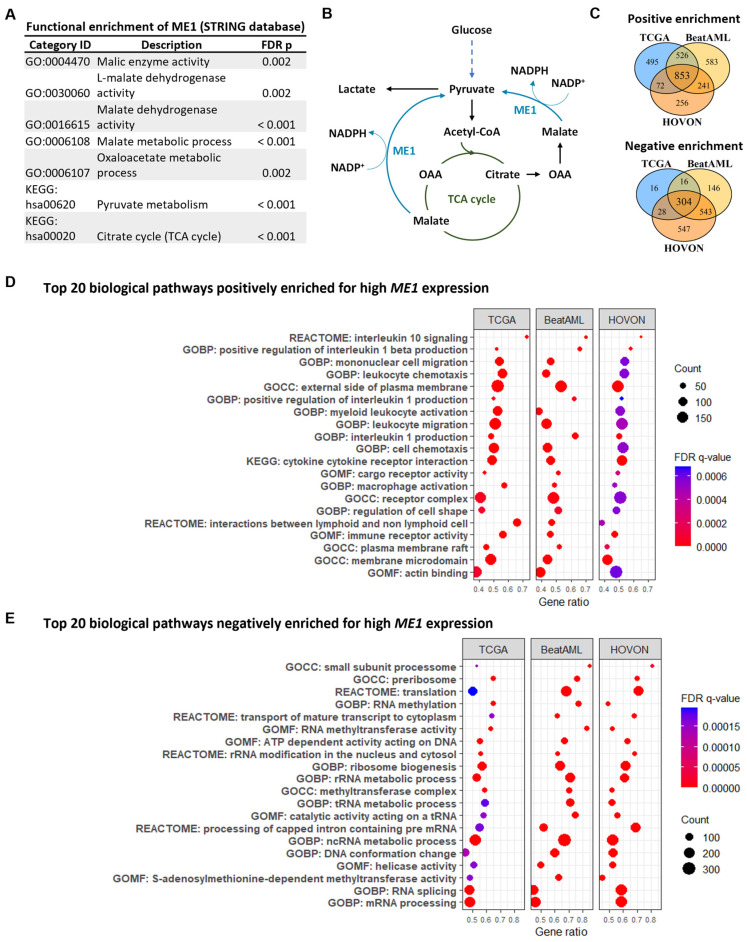
High *ME1* expression reflects relevant biological and immunological signatures. (**A**,**B**) STRING analysis validating ME1 as cytosolic enzyme necessary for replenishing pyruvate from malate, a process that generates NADPH. (**C**) Venn diagram showing the number of gene sets robustly enriched with *ME1* expression. (**D**) Top 20 biological pathways positively and (**E**) negatively correlated with *ME1* expression.

**Figure 5 cancers-15-00296-f005:**
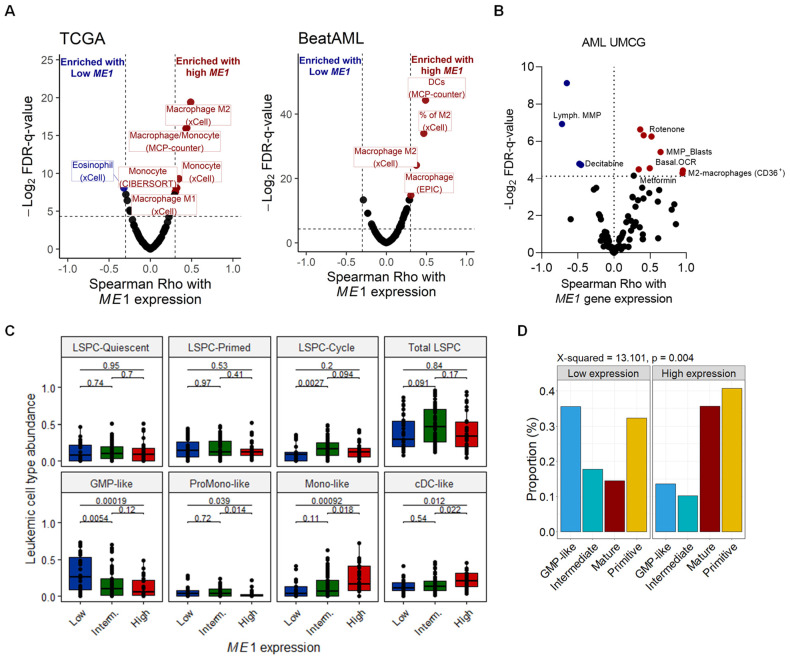
Estimation of cell type abundances according to *ME1* expression in AML. (**A**) *ME1* expression predicts increased M2 macrophages infiltrations in the bone marrow microenvironment of AML. (**B**) High *ME1* expression correlated with CD36 abundance in primary AML samples. (**C**) Deconvolution of AML samples to predict AML hierarchy subtypes demonstrated that *ME1* expression was associated with mono-like clones. Patients were divided in three groups according to their *ME1* expression levels, whereby low, intermediate, and high groups correspond to quartile 1, quartile 2 + 3, and quartile 4 of *ME1* expression, respectively. (**D**) Mature AML subtype is frequent in high *ME1* expression cells, while GMP is associated with low *ME1* expression.

**Figure 6 cancers-15-00296-f006:**
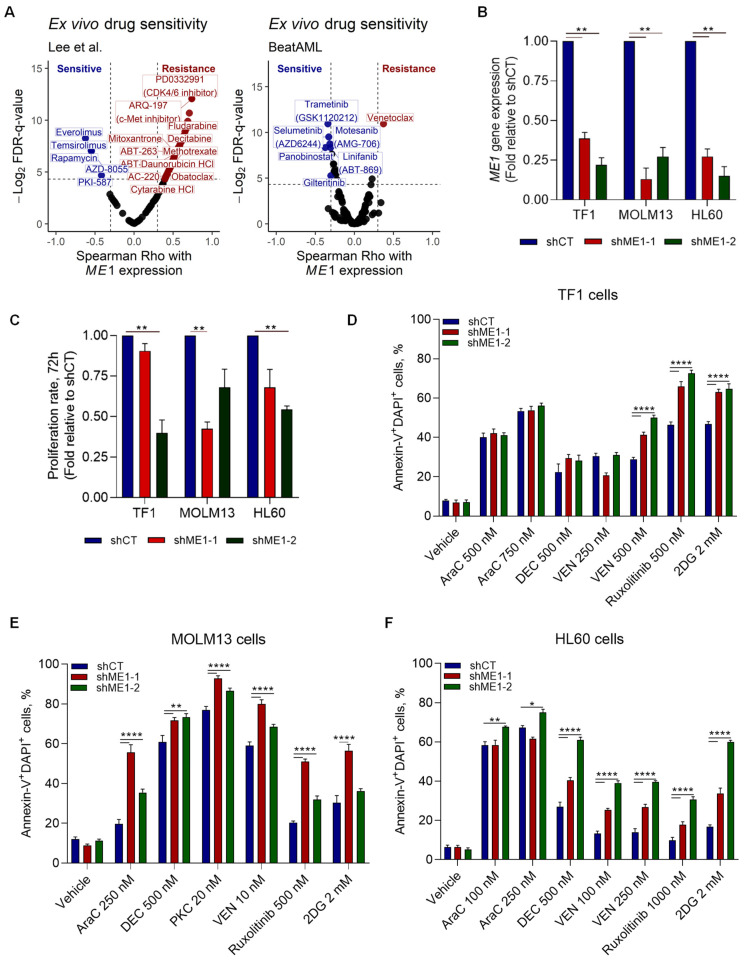
Drug sensitivity according *ME1* expression in AML. (**A**) Ex vivo drug sensitivity indicated drug responses according to *ME1* expression. (**B**) Knock-down of ME1 expression (shME1-1 and shME1-2) in AML cell lines. (**C**) Proliferation rates of sh*ME1* cells in comparison with control (shCT cells). (**D**–**F**) Treatment of sh*ME1* cells with different drugs confirming cell vulnerabilities against standard drugs used in AML. * *p* ≤ 0.05; ** *p* ≤ 0.01; **** *p* ≤ 0.0001.

## Data Availability

No new data were created or analyzed in this study. Data sharing is not applicable to this article.

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
