# Peer review of "High ME1 Expression Is a Molecular Predictor of Post-Transplant Survival of Patients with Acute Myeloid Leukemia"

_cancers, 2022, doi:10.3390/cancers15010296_

Round 1

Reviewer 1 Report

-     Article highlighted an interesting and fairly novel finding of the prognostic significance of ME1 in AML post transplant using 3 different databases. However, there are several comments below that the author(s) can address to make the paper stronger / more presentable.

 -  -   What is the rationale of selecting and presenting findings from all 3 databases (TCGA, BeatAML, HOVON)? Is there a reason only one database (e..g BeatAML) and not the others was presented in some cases e.g. in the drug sensitivity study on Figure 5D?  I wonder if the paper would look more presentable if the key findings from the 3 databases are summarized in a figure instead of 3 long tables, and other findings can be in supplementary tables and similarities/differences are highlighted in text.

-        -  All 3 databases seem to use ELN based classification. It would be more interesting if there’s comparison between different classification e.g. ELN vs NCCN.

-       - What are the high and low gene expression defined as? Is there a cutoff value or VAF to distinguish between these expression levels?

        - It was observed that high ME1 expression was associated with shorter overall survival in patients treated with chemotherapy alone in the BeatAML and HOVON cohorts. However, these findings were not consistent, as this difference in survival was not identified in the TCGA cohort (Figure 3E). Can you expand the discussion on what may explain this difference, as well as other differences observed between the databases?

-        - Introduction could have some more work e.g. describing what ME1 is in the intro rather than other sections and previous works on ME1 in other tumor types.

-

Author Response

We would like to thank the reviewer for these comments. Relevant points were indicated, helping us to improve our work. Here our replies:

  1. The main rationale why we chose the TCGA, BeatAML and HOVON cohorts was the availability of HSCT status, which was unavailable in other datasets. Additionally, these three cohorts, are well known and representative cohorts for AML, permitting us to draw reliable conclusions and allowing future readers to easily validate our findings when necessary.

With respect to of why we used not all three for some analysis, is because the data needed to perform those analysis is available only for some datasets. Specifically, the ex vivo drug sensitivity analysis was available only for the BeatAML cohort and for this reason  we added an additional public dataset (Lee et al.) comprehending an AML drug screen. However, the later did not include any the clinical data fo r survival analysis based on ME1 expression. In our new version, we included the drug sensitivity test for AML cell lines  silenced for the ME1 gene (Figure 6). As mentioned in the article, we confirmed the sensitivity for several drugs in relation with ME1 expression.

Tables with clinical descriptions were moved to the supplementary section (Supplementary Tables S3, S4 and S5).

  1. About the use of NCCN in addition to ELN. Unfortunately, NCCN consensus risk classification is not available for any of the three datasets. The only risk stratification that we could use to compare, would be the Medical Research Council (MRC) classification (doi.org/10.1182/blood-2019-128234), but the ELN-risk stratification already comprehends most of the MRC features, so we kindly asked the reviewer to not include that one, to avoid overlapping conclusions.
  2. Regarding the cutoff used to define high and low ME1 As described in the method section (statistical analysis), we calculated the optimal cutpoint based on ME1 expression and the survival data, by using the ROC curve. Since ME1 was not mutated in the AML cohorts that we included, we did not consider using any other genetic parameter (such as the VAF), to help the cutpoint selection.
  3. To elaborate on the difference of the ME1 prognosis prediction between patients treated with chemotherapy and HSCT, it is important to mention that the rationale of our statistical strategy for gene filtering was to find genes strongly associated with survival specific for after HSCT. Consequently, it was not unexpected that the ME1 gene could also predict survival in the context of 3+7 schemes used for induction before HSCT. Here, we did not find consistent association between ME1 expression and overall survival when we considered all AML cases or patients treated with chemotherapy only (only two cohorts from three resulted in this association). However, in our validation cohort we also find this association (Supplementary Figure S1B). Although the exact reasons of why ME1 is stronger in predicts survival after HSCT are not dipped in this work, we tried to provide some functional studies to explain the clinical consequences of high ME1 expression. For instance, we demonstrated that patients with high ME1 expression also presented high abundance of M2-macrophages (validated in our own samples in the current version, Figure 5B), increased resistance to certain AML drugs  (Figure 6) and are associated with a metabolism driven by oxidative phosphorylation (Supplementary Figure S5A-B). Altogether, these data suggest that AMLs with high ME1 expression are characterized by a supportive tumor microenvironment, increased resistance to drug induced apoptosis and a more OXPHOS-like metabolism, all factors potentially contributing to poor outcome of ME1high
  4. We thank the reviewer for this comment. As the main objective of this work was to find genes associated with survival after HSCT, we decided focusing our introduction on the problematic about biomarkers for HSCT and leave relevant information about ME1 in the discussion section.

Reviewer 2 Report

Authors declared the significance of ME1 expression as a predictor of port-trans-plant survival of patients with AML by using published data cohort. They should validate their results by using author's original cohort so as to confirm the significance of ME1. 

Reviewer thinks there is critical points in methodology. Authors analyzed only database cohorts, which are collected according to different protocol. When researchers use those results, they should consider potential biases. They must confirm the same results by using original cohort, which could erase the potential bias.

Author Response

We thank the reviewer for the  comments. In this revised version, we included our own cohort to perform some additional analysis. This cohort correspond to patients from the University Medical Center of Groningen (UMCG, n= 80 patients – of which 32 received 3+7 induction scheme). With this cohort we validated the ME1 expression as a prognostic marker in AML (n=32) (Supplementary Figure S1B-C). Unfortunately, due to the limited number of patients treated with HSCT (n=20), we were unable to perform analysis on transplant outcomes based on ME1 levels. Nevertheless, the advantage of this cohort is the fact that all the included patients were fully functionally characterized regarding the surface marker expression distinguishing different cellular subpopulations (e.g., M2-like macrophages) (Figure 5), ex vivo drug sensitivity (Figure 6), and functional respiration capacity (measured by seahorse analysis) (Supplementary Figure S5). These data allowed us to validate our previous findings , like the differential drug sensitivity and the abundance of M2-like macrophages. Additionally, in the current version we included three AML cell lines genetically modified using lentiviral system to reduce the ME1 gene expression and we investigated some of the described features associated with ME1 high AML like drug sensitivity (Figure 6) and their metabolic state (Supplementary Figure S5). Therefore, in the revised version we validated some results, reinforcing the concept of ME1 as a predictor of survival after HSCT.

Reviewer 3 Report

Reviewer’s comments

Title: High ME1 expression is a molecular predictor of post-transplant survival of patients with acute myeloid leukemia

The authors determined that high MEI expression is associated with poor outcome of adult AML patients who received hematopoietic stem cell transplantation (HSCT). They also try to explain why AML patients with high ME1 expression are resistant to chemotherapy followed by HSCT. Interestingly, high ME1 expression does not seem to be related to any other known risk factors, although the patients’ number in this study is small. Although the manuscript contains potentially interesting information, several issues should be resolved.   

. 

(Major concerns)

1.     As the authors described in this manuscript, high expression of ME1 is also associated with poor prognosis of adult AML patients who did not receive HSCT in two cohorts (presented in Fig. 3E). Thus, the reviewer speculates that disease status of the patients with high ME1 expression might affect the outcome of HSCT. Do you have any information about disease status prior to HSCT according to the expression level of ME1 in these patients?

2.     It is likely that high expression of ME1 is associated with chemo-resistance of AML cells. Thus, the reviewer thinks that the authors should evaluate whether high expression of ME1 is associated with outcome of adult AML patients irrespective of receiving HSCT. It might be also interesting whether high expression of ME1 show the different effect on outcome according to the ELN risk. The reviewer could not see any benefit to select the patients receiving HSCT.

(Minor concerns)

1.     MLL should be KMT2A thought the manuscript.

Author Response

Thank you for these comments. Important points were indicated, helping us to improve our work. Here our replies:

  1. We thank the reviewer for the We included an analysis to evaluate the possible association between the response to induction chemotherapy and the ME1 expression. For this analysis we used the BeatAML cohort (the only cohort with this data available), considering three groups of patients: total AML patients, only chemotherapy treated patients and patients that receive HSCT. As shown in Supplementary Figure S1A, we did not find an association between the disease status and ME1 expression.
  2. In regard to the application of ME1 in HSCT patients, and why it is not applied in all AML cases. In the new version of the manuscript, we now included in Figure 3F an univariate Cox regression analysis showing the outcome of   all AML patients, chemotherapy treated only, and patients treated with HSCT based on ME1 expression . In this analysis we found that ME1 gene expression is a better predictor of prognosis when only HSCT-treated patients are considered. Interestingly, in our own cohort, which includes both patients treated with HSCT and chemotherapy only, ME1 expression could stratify the patients regarding their overall survival Supplementary Figure S1B.

Reviewer 4 Report

In this manuscript, Rojas et al found that ME1 was associated with shorter survival rates in AML patients submitted to HSCT after analyzing three publicly available adult AML cohorts’ datasets (TCGA, BeatAML and HOVON) using bioinformatic tools. Additionally, the authors analyzed biological pathways, microenvironmental cellular signatures, AML blasts subtypes, and ex vivo drug sensitivity associated with relevant genes for prognosis. Particularly, the authors identified ME1 expression associated with M2 macrophage-enriched microenvironment composition, and with primitive and mature AML blasts hierarchy. Their findings are very interesting and support ME1 as a biomarker of poor response to HSCT in AML. The manuscript was well written, and the data were nicely presented in tables and figures.  

Additional comments:

1.     Is ME1 associated with shorter survival rates in AML patients submitted to HSCT in other cohorts besides TCGA, BeatAML and HOVON?

2.     Is ME1 associated with shorter survival rates in other cancer patients submitted to HSCT besides AML?

3.     Is ME2 or ME3 also associated with shorter survival rates in AML patients submitted to HSCT? can the authors explain why ME2 or ME3 did not come out from the analysis?

4.     In figure 5A, can the authors explain why macrophages did not show up in the cell type abundance of HOVON figure?

5.     In figure 5E, the authors found that high ME1 expression was correlated with resistance to histone deacetylase inhibitors (HDACi) and to ruxolitinib (JAK1/2 inhibitor) but it responded better to CDK9_5038, a CDK9 inhibitor, and ERK_6604, an ERK1/2 inhibitor. Can the authors use experimental methods, at least using AML cell lines, to verify and confirm these findings?

6. Are there any IHC staining data available to show higher macrophage markers such as CD11b, CD68 or CD163 in ME1-high AML samples compared to ME1-low samples?

Author Response

We thank the reviewer for the positive feedback and comments. Here our replies:

  1. In this new version of the manuscript, we included our own validation cohort of AML patients (n=32) (Supplementary Figure S1B-C). Irrespective of the administered treatment, the survival of AML patients could be stratified based on ME1 expression . Although 20/32 patients were submitted to HSCT, we couldn’t add a fourth representative cohort to validate ME1 expression in the context of HSCT. Nonetheless, in this work we used three representative cohorts of AML and found the same results in an independent analysis. Therefore, ME1 is presented here as a new biomarker for HSCT-treated AML patients.
  2. Respect to the second question. In this work we focused on AML. For this reason, we decided not to include any analysis on any other hematological malignancies, where  HSCT would represent a curative option (e.g., myelodysplastic syndrome and myeloproliferative neoplasm). We hope that future studies will focus on evaluating the role of ME1 expression in other tumors.
  3. About ME2 and ME3. In our work, we performed a whole transcriptome analysis, and after applying our statistical filters, ME2 and ME3 genes were not associated with prognosis prediction in AML patients submitted to HSCT (Supplementary Table S1 of the first version of the article). Biologically, ME2 and ME3 are mitochondrial enzymes, while ME1 is cytosolic. Perhaps, this and other biological differences explain why ME2 and ME3 are not highlighted in this work.
  4. About figure 5A. We are grateful for this comment. We consider that technical difference between HOVON (microarray) versus TCGA and BeatAML vs. (RNA-sequencing) could explain the result presented in Figure 5E of the previous version of our manuscript. These analyses were now updated in the current version (Figure 5). Furthermore, to validate the increased frequency of M2-macrophages in ME1high AMLs, we included the analysis of samples present in our own cohort (UMCG) and found that M2-like macrophages are in fact increased in abundance when the expression of ME1 is high (Figure 5B).
  5. With respect to the drug sensitivity, we performed experiments with three AML cell lines, in which the ME1 was silenced , and the sensitivity for some of the previously pointed drugs were tested (Figure 6 of the current version). Finally, we confirmed the association between ME1 expression and sensitivity to standard drugs used for
  6. Finally, in the current version, we analyzed the macrophage landscape of a group of AML patients harboring low and high ME1 expression, using flow cytometry-based studies. As a result, we confirm that M2 macrophages are more abundant when the AML blast cells have a higher ME1 expression (Figure 5B of the current version).

Reviewer 5 Report

English should be revised. 

Author Response

Thanks for your comment. The English language was revised in the current version.

Round 2

Reviewer 3 Report

The manuscript has been revised sufficiently based on the reviewer's comments.